# Factors Associated with Aspirin Resistance in Hong Kong Chinese Patients with Stable Coronary Heart Disease Using the Multiplate^®^ Analyzer and Serum Thromboxane B_2_

**DOI:** 10.3390/pharmaceutics14102099

**Published:** 2022-10-01

**Authors:** Weiwei Zeng, Tanya T. W. Chu, Elaine Y. K. Chow, Miao Hu, Benny S. P. Fok, Juliana C. N. Chan, Bryan P. Y. Yan, Brian Tomlinson

**Affiliations:** 1The Second People’s Hospital of Longgang District, Shenzhen 518112, China; 2Department of Medicine and Therapeutics, The Chinese University of Hong Kong, Shatin, Hong Kong SAR 999077, China; 3Faculty of Medicine, Macau University of Science & Technology, Macau 999078, China

**Keywords:** aspirin resistance, platelet aggregometry, thromboxane B_2_, white blood count

## Abstract

*Introduction:* Aspirin resistance may be associated with various conditions. We measured serum thromboxane B_2_ (TXB_2_) and platelet function using the Multiplate^®^ Analyzer with arachidonic acid (ASPI) in patients on long term aspirin therapy to identify aspirin resistance and associated factors. *Materials and Methods:* Chinese patients with stable coronary heart disease had samples for serum TXB_2_ and ASPI measurement taken before and 1 h after taking a morning dose of 80 mg aspirin. *Results:* In 266 patients with mean age 66.6 ± 10.7 years, 17% were female and 55% were current or previous smokers. TXB_2_ and ASPI measurements were significantly higher before the dose than at 1 h post dose, with 46% of subjects having high ASPI values (AUC > 300 AU*min) pre dose compared with 27% at 1 h post dose. TXB_2_ and ASPI measures of platelet aggregation showed weak correlations, which were only significant before the dose (r = 0.219, *p* = 0.001). Increased ASPI measurements were associated with white blood cell (WBC) count, haematocrit, platelet count and heart rate at 24 h post dose but only with WBC count, smoking history and heart rate at 1 h post dose. Diabetes was not associated with reduced platelet response to aspirin. The WBC count associated with aspirin resistance was over 6.55 × 10^9^/L by receiver operating characteristic analysis. *Conclusions:* The antiplatelet response to aspirin was reduced in a large proportion of patients. Patients with higher WBC count within the normal range appear to be at increased risk of aspirin resistance. Higher or more frequent doses of aspirin may be needed in many patients.

## 1. Introduction

Aspirin resistance, also known as high on-aspirin residual platelet reactivity (RPR), can be defined as the inability of aspirin to adequately prevent the production of platelet cyclooxygenase-1 (COX-1)-derived thromboxane A_2_ (TXA_2_) and it can be detected by laboratory tests of platelet TXA_2_ production by measurement of metabolites such as serum thromboxane B_2_ (TXB_2_) or platelet function testing that depends on platelet thromboxane production [1,2,3]. Laboratory aspirin resistance has been estimated to occur in 5–60% of patients on aspirin treatment, depending on the type of patients studied, different study designs, different tests of platelet function, and different definitions of aspirin resistance [1,4,5,6,7,8].

Despite aspirin therapy, a considerable number of patients, particularly those with diabetes, continue to have atherothrombotic events, which is referred to as aspirin treatment failure [9,10]. Although the aetiology of aspirin treatment failure is multifaceted, it may be attributable in part to the high on-aspirin RPR or aspirin resistance [1,11,12].

TXB_2_ is the major metabolite of platelet TXA_2_ and it is derived only from platelet TXA_2_, and thus serum TXB_2_ has been suggested to be the most specific test to measure the pharmacological effect of aspirin on platelets [13]. There was a moderate correlation between the serum TXB_2_ assay and other assays measuring platelet function and inadequate aspirin response [14,15]. However, platelets may be activated through multiple pathways other than TXA_2_, and thus using this method to measure platelet response to aspirin may underestimate the prevalence of aspirin resistance or “aspirin non-responders”.

Platelet function ex vivo has been measured in patients on aspirin treatment by several functional assays including light transmission aggregometry, whole blood aggregometry, Multiplate^®^ Analyzer, Platelet Function Analyzer-100^®^, VerifyNow Aspirin^®^, urinary 11-dehydro-TXB_2_ and serum TXB_2_ measurement, etc. However, these functional assays have variable specificity for measuring the effects of aspirin on platelet function and there is a poor correlation between results from different tests [16,17,18]. The Multiplate^®^ device performs multiple electrode impedance aggregometry in whole blood, the Platelet Function Analyzer-100^®^ measures the time required for platelet aggregation to occlude an aperture, whereas the VerifyNow^®^ measures platelet aggregation in whole blood based on change in light transmission [19]. The Multiplate impedance platelet aggregometry induced by arachidonic acid (ASPI) measurement has been used in many studies and in participants taking aspirin the highest platelet response variations with arachidonic acid were found in blood samples anticoagulated with hirudin [20,21]. A randomized clinical trial in healthy subjects treated with aspirin or clopidogrel suggested that any one of these three point-of-care devices can be used for assessing the treatment effects of aspirin or clopidogrel on platelet aggregation at the study population level [19].

This study compared the ASPI measurement for the Multiplate^®^ device with the measurement of serum TXB_2_ as the reference standard and examined the association of phenotypic factors on the antiplatelet effect of aspirin, and the prevalence of aspirin resistance in Hong Kong Chinese patients with stable coronary heart disease (CHD). 

## 2. Material and Methods

### 2.1. Study Population

Eligible patients aged ≥18 years with CHD receiving long-term mono-antiplatelet therapy for ≥3 months with plain, uncoated aspirin (80 mg once daily) were invited to participate in the study in the outpatient clinic at the Prince of Wales Hospital, Hong Kong. The study was performed in accordance with Good Clinical Practice (GCP) and the Principles of the Declaration of Helsinki. The study was approved by the Joint Chinese University of Hong Kong–New Territories East Cluster Clinical Research Ethics Committee with reference number CRE-2014.516-T. Written informed consent was obtained at the time of enrolment from all subjects. This study was reported with respect to the recommendation for reporting randomized clinical trials as the statement of Consolidated Standards of Reporting Randomized Clinical Trials (CONSORT) in this article. 

Patients had previously been diagnosed with CHD based on the following criteria: ischaemic electrocardiogram (ECG), positive exercise test, symptoms of angina pectoris, history of acute coronary syndromes, coronary angioplasty, or coronary artery bypass surgery. Patients who had a myocardial infarction, stroke, coronary artery bypass surgery or other revascularization treatment, unstable angina or angioplasty during the previous 3 months were excluded from the study. Patients on ticlopidine, prasugrel, clopidogrel, dipyridamole or other antiplatelet or anticoagulant medications were not recruited. Patients on regular therapy with anti-inflammatory drugs or other drugs containing aspirin or non-steroidal anti-inflammatory drugs (NSAIDs), or traditional Chinese medicine that have direct effects on the haemostasis were excluded from the study unless they were willing to discontinue these treatments for at least 2 weeks prior to study participation.

The patients were instructed to take aspirin in the morning for at least 7 days before the research visit, if they were not already doing so. Utilizing standard equipment and methodology, anthropometric measurements of body weight and height were obtained. The percentage of total body fat was measured using an impedance device (TANITA Body Composition Analyzer BF-350, TANITA Corporation, Tokyo, Japan). The sitting blood pressure and heart rate was taken using a semiautomatic sphygmomanometer (Critikon Dinamap 8100; GE Medical Systems Information Technologies, Louisville, KY, USA) after resting seated for 5 min. 

During the study visit, blood samples were obtained in the morning before the aspirin dose and one hour after the dose to measure the trough and peak levels of the tests for the anti-platelet effects of aspirin. In the study centre, drug adherence was evaluated face-to-face via a personal interview. Demographic information, medical history and recent laboratory test results were collected. 

### 2.2. Sample Collection

Participants arrived at the study centre after fasting for at least 10 h. Patients’ fasting blood samples were taken immediately before and one hour after a single 80 mg aspirin dose. In a serum separator tube, a 5 mL blood sample was collected and allowed to clot at room temperature before centrifugation at 4 °C. The separated serum was stored at −80 °C in aliquots until TXB_2_ analysis. A 3 mL blood sample was collected in a hirudin blood tube for the platelet aggregation test with the Multiplate^®^ analyzer, which was performed within 3 h of blood collection. During the study, a total of 16 mL blood was taken from each participant.

### 2.3. Multiplate^®^ Analyzer

The platelet activity of the samples was measured with the hirudin blood using the Multiplate^®^ Analyzer (Roche Diagnostics International Ltd., CH-6343 Rotkreuz, Switzerland) according to the manufacturer’s instructions for the ASPI test. It was analysed within 0.5–3 h after blood collection. Platelet aggregation is measured continuously using electrical impedance for 5 min after addition of 20 µL of reconstituted ASPI reagent containing 0.5 mM arachidonic acid at final concentration to the test cells in channel one and channel two, which serves as a paired control. The impedance is converted to arbitrary aggregation units (AU) and plotted over time, and the area under the aggregation curve (AUC) was calculated as AU × minutes (AU*min).

### 2.4. Serum Thromboxane B_2_ Assay

TXB_2_ was detected in the serum of 253 patients using Cayman enzyme immunoassay kits (TXB_2_ EIA Kit Item no. 501020, Cayman Chemical Company, Ann Arbor, MI, USA) with a detection range of 1.6–1000 pg/mL. All the calibration standards and buffers/reagents for the test were prepared in accordance with the kit’s instructions. All samples were diluted in a 1:1 ratio with EIA buffer. The standards were assayed in duplicate, whereas the samples were assayed singly. The data of logit (B/B_0_) vs log concentrations of standards was plotted and a linear regression fit performed, where Logit (B/B_0_) = ln [(B/B_0_)/(1-B/B_0_)], with B_0_ representing maximum binding and B representing sample/standard binding. Each sample concentration was determined by the equation obtained from the standard curve. 

### 2.5. Other Assays

Other biochemical assays for glucose, renal and liver function and lipids were measured on a Roche Modular Analytics system (Roche Diagnostics GmbH, Mannheim, Germany) using standard reagent kits supplied by the manufacturer of the analyser. The analytical performance of these assays was within the manufacturer’s specifications. Low-density lipoprotein cholesterol level was estimated by using the Friedewald formula. 

### 2.6. Haematology Assays

Complete blood profile, including white blood cell (WBC) count, was measured using an automated cell counter (GEN-S; Beckman Coulter, Miami, FL, USA).

### 2.7. Definition of Aspirin Resistance with the ASPI Test

In this study, an ASPI AUC value >300 AU*min was used as the conventional cut-off to identify aspirin resistance in patients taking 80 mg aspirin daily [22]. 

### 2.8. Statistical Analysis

A sample size calculation based on previous similar studies estimated that obtaining results for both tests of anti-platelet activity in 250 patients would allow the study to have at least 80% power to detect a correlation coefficient of at least 0.2 between the two measurements with level of significance of 0.05 in patients with CHD. The study was not powered for validation of a clinical prediction model [23]. Correlations between the results obtained with the two assays, irrespective of aspirin resistance classification, were assessed by the Pearson’s correlation coefficient. The strength of the correlation (r-value) was interpreted according to the following common definitions: 0.00 to 0.19 “very weak”, 0.20 to 0.39 “weak”, 0.40 to 0.59 “moderate”, 0.60 to 0.79 “strong”, and 0.80 to 1.0 “very strong” [24]. Receiver operating characteristic (ROC) analysis was performed comparing the two tests and factors which may influence them. The relationships between baseline demographic factors and aggregation on the ASPI AUC (AU*min) measurement before and after the aspirin dose were analysed by multiple stepwise linear regression entering platelet count, haematocrit (Hct), WBC count, fasting blood glucose (FBGL), body fat percentage (B fat%), systolic blood pressure (SBP), diastolic blood pressure, heart rate, waist hip ratio (WHR), body mass index (BMI), having DM and cigarette smoking into the equation. Simultaneously, normal distribution and collinearity of explanation variables were detected. Data were analysed using SPSS version 17.0 (SPSS Inc., Chicago, IL, USA).

## 3. Results

A total of 1899 potentially eligible patients’ medical records were reviewed, and 471 were contacted, with 267 agreeing to participate in the study. In 255 patients, data for the Multiplate^®^ impedance platelet aggregometry aspirin (ASPI) assessments before and 1 h after the aspirin dose were available (Figure 1). Only 251 patients had serum TXB_2_ levels measured. Table 1 and Table 2 presented clinical, demographic, and laboratory information for all patients as well as subjects with and without diabetes mellitus (DM). 

The mean (± SD) age was 66.6 ± 10.7 years, the mean body mass index was 25.1 ± 3.5 kg/m^2^, 17% were female and 55% were current or previous smokers. There were no non-compliant patients based on pill counting and face-to-face interviews. The suppressed levels of TXB_2_ (mean ± SD, 91.3 ± 90.7 pg/mL pre dose) further confirmed that all patients were compliant with aspirin in this study.

Compared with subjects without DM, those with DM were more likely to have hypertension, peripheral artery disease (PAD) and previous stroke, and to be treated with angiotensin converting enzyme inhibitors (ACEIs) or angiotensin receptor blockers (ARBs) (Table 1 and Table 2). Patients with DM were older and had a greater waist circumference and WHR, greater SBP, fasting glucose, triglycerides and lower levels of high-density lipoprotein cholesterol (HDL-C), low-density lipoprotein cholesterol (LDL-C), red blood cell count (RBC), haemoglobin (Hb) and Hct. Platelet count was not different between those with and without DM. 

There were no significant differences in the serum TXB_2_ levels and the ASPI measures between patients with and without DM (Table 3). The serum TXB_2_ levels and the ASPI AUC measures were both significantly higher in the samples pre dose, compared with the samples 1 h after the dose of aspirin (Table 3). The mean values before and 1 h after aspirin for the ASPI test for all patients were 311.8 ± 167.5 AUC and 260.8 ± 122.1 AUC, respectively (*p* < 0.001). A similar pattern was observed in the DM and non-DM groups. There was no significant relationship between platelet aggregation and age, body mass index, body fat percentage and fasting glucose levels.

There were weak correlations between the serum TXB_2_ levels and the ASPI measure of platelet aggregation which were significant (*p* = 0.001) before but not after the dose (r = 0.219 and 0.118, respectively, Figure 2 and Figure 3).

In total, there were 120 (46%) subjects with ASPI values >300 before the aspirin dose, including 75 (48%) from the non-DM group and 45 (41%) from the DM group. After the aspirin dose there were a total of 71 (27%) subjects with ASPI values >300, including 45 (29%) from the non-DM group and 26 (25%) from the DM group. There were 60 (22.6%) subjects with ASPI values >300 before the dose that became <300 after the dose of aspirin, including 34 (21.8%) from the non-DM group and 26 (23.9%) from the DM group, without any significant difference between the DM and the non-DM groups. There were 12 (4.5%) subjects with values <300 before the dose that became >300 after the dose, 4 (2.6%) from the non-DM group and 8 (7.3%) from the DM group, without any significant difference between groups. There were 10 subjects with missing values for one of the two ASPI AUC values, either pre dose or post dose.

The regression diagnostics shows that the model studentized residuals of histogram followed a normal distribution (Std.Dev = 0.992) and none of the included variables had collinearity (VIF < 10). The baseline demographic factors associated with increased aggregation on the ASPI measurement 24 h post dose (r = 0.481, adjusted r square = 0.219, *p* < 0.0001) were WBC count (B = 34.725, *p* < 0.0001), Hct (B = 772.905, *p* = 0.001), platelet count (B = 0.426, *p* = 0.026) and heart rate (B = 2.629, *p* = 0.019) (Table 4).

On the ASPI measurement 1 h post dose, only the WBC count (B = 21.73, *p* < 0.0001), smoking history (B = 43.894, *p* = 0.002) and heart rate (B = 1.918, *p* = 0.026) were significantly associated (Table 5). The regression diagnostics also shows that the model studentized residuals of histogram followed a normal distribution (Std.Dev = 0.996) and none of the included variables had collinearity (VIF < 10). 

WBC count, Hct and platelet count were significantly associated with ASPI AUC values >300 prior to aspirin dosing based on receiver operating characteristic (ROC) analysis ASPI AUC values >300 pre dose with WBC count having the most significant relationship (Figure 4).

A cut-off of 6.55 × 10^9^/L for WBC count was most associated with aspirin resistance (area under the ROC curve: 0.679, *p* < 0.0001, *n* = 255; sensitivity: 0.625; specificity: 0.669).

The TXB_2_ level associated with aspirin resistance identified by ASPI AUC values >300 on ROC analysis was of borderline significance (*p* = 0.037) with low values for sensitivity and specificity (sensitivity: 0.456; specificity: 0.451) and a cut-off of 44.6 pg/mL for serum TXB_2_ to indicate aspirin resistance (Figure 5).

A cut-off of 44.6 pg/mL for serum TXB_2_ was best associated with aspirin resistance with the area under the ROC curve of 0.488 (area under receiver operating characteristic: 0.488, *p* = 0.037, *n* = 247; sensitivity: 0.456; specificity: 0.451). 

## 4. Discussion

In this study, at 24 h post dose, 46% of patients taking aspirin 80 mg daily were found to have ASPI AUC values > 300 AU*min, which is the conventional cut-off to identify aspirin resistance [22,24]. The existence and significance of aspirin resistance has been debated but aspirin resistance determined in certain ways has been associated with an increased risk of major adverse CHD events in patients with stable CHD in several studies [25,26,27].

The measurements of serum TXB_2_ showed a very weak correlation with the ASPI AUC values and from the ROC analysis, it appears that serum TXB_2_ would not be useful to identify patients with aspirin resistance identified by the ASPI AUC. In some previous studies, the measurements of serum TXB_2_ have been used to identify compliance with aspirin treatment rather than to identify aspirin resistance [11,28].

Having type 2 DM has been identified as a risk factor for aspirin resistance in several previous studies [29]. In one large study of 900 patients with stable CHD, type 2 DM and BMI were independent determinants of increased arachidonic acid (AA)-induced platelet aggregation with the Multiplate^®^ Analyzer using citrate and hirudin anticoagulated blood and also with the VerifyNow^®^ Aspirin Assay [30]. Other factors associated with reduced aspirin effect included platelet count, previous myocardial infarction, previous coronary artery bypass grafting, age, smoking and female gender [30]. We did not find a relationship between having type 2 DM and reduced aspirin response in the present study. This could be due to a number of factors including the relatively good glycaemic control in the patients with DM, as shown by the fasting plasma glucose levels. Glycosylated Hb (HbA1c) levels were not measured in this study visit but had been considered satisfactory at previous clinic visits. There was also frequent use of other protective treatments, such as statins and ACEIs or ARBs, in the patients with DM.

The relationships between reduced platelet response to aspirin and WBC count, platelet count, and RBC or Hct have been described in several previous studies [31]. In Chinese patients with acute ischaemic stroke, aspirin resistance evaluated by the PFA-100 test was associated with WBC count (r = −0.125, *p* = 0.041) and LDL-C and predicted an unfavourable stroke outcome at 90 days [32]. In a study in Turkey, patients with stable angina with aspirin resistance had significantly higher WBC count, neutrophil counts, neutrophil-to-lymphocyte ratios, serum uric acid (SUA) levels, high-sensitivity C-reactive protein levels, and fasting blood glucose levels and high SUA levels were an independent predictor of aspirin resistance after multivariate analysis, with a cut-off value of 6.45 mg/dl for SUA to predict aspirin resistance with 79% sensitivity and 65% specificity on ROC analysis [33]. In another study, in patients with acute coronary syndromes, the anti-aggregant effect of aspirin assessed by PFA-100 closure times with ADP (PFA/ADP) and epinephrine (PFA/EPI) were modulated not only by the direct action on platelets, but also by erythrocyte deformability and WBC count [34].

We have demonstrated that WBC count and Hct level were associated with the response to aspirin treatment, which has been indicated by other studies showing the interaction of aggregation impedance and other cellular components [35,36]. We also found an association between aspirin resistance and heart rate. A previous study found that heart rate variability was decreased, and sympathetic activity increased in patients with aspirin resistance and stable CHD [37]. It is plausible that increased sympathetic activity associated with increased heart rate will result in platelet activation. Aspirin resistance has also been associated with hypertension [38,39] but we did not find any association with systolic and diastolic blood pressures in this study. As with diabetes, the treatments used for hypertension may interact with the effect of aspirin on platelets.

Increased platelet count, as in essential thrombocythaemia, has been associated with aspirin resistance and this may be explained by accelerated platelet turnover [40]. Increase platelet turnover and oxidative stress have been suggested to be mechanisms of aspirin resistance with DM in some studies [41]. The mechanism of the association of smoking with aspirin resistance may also be through oxidative stress. The mechanism of the association of higher WBC count and aspirin resistance may be related to aspirin insensitive thromboxane generation as monocytes may generate TXA_2_ from increased COX-2 expression in inflammatory conditions [31,42]. The mechanism of the association of higher Hct and aspirin resistance remains unclear, but erythrocyte deformability may be relevant [34]. 

It would be interesting to follow-up the patients in this study to determine if any of the measurements made influence the cardiovascular outcome, but this was beyond the scope of the study. Further studies would be useful to determine whether increasing the dose or the dosing frequency of aspirin could improve the measure of platelet aggregation. A large clinical trial would be necessary to prove that such action would be beneficial and that would be very difficult to perform.

This study has several limitations. We only used the ASPI test with the Multiplate^®^ Analyzer and using other agonists such as collage, thrombin or ADP may have given different results. Furthermore, ex vivo platelet function tests may not accurately reflect in vivo conditions. We did not measure the differential WBC count, which might have provided more useful information. The sample size of the study was relatively small and would not be sufficient to provide validation of a clinical prediction model. A recent review of this topic concluded that external validation of a clinical prediction model would require a sample size of several thousand participants [23]. 

## 5. Conclusions

In the Chinese patients with CHD who participated in this study, 46% showed evidence of aspirin resistance by the Multiplate^®^ Analyzer ASPI test 24 h after the last dose of aspirin and 27% showed aspirin resistance at 1 h post dose. The measurements of serum TXB_2_ did not correlate well with the ASPI test and are unlikely to be useful in identifying patients with aspirin resistance. Aspirin resistance was not associated with having DM in this study but did show associations with WBC count, Hct, platelet count, smoking history and heart rate. The WBC count showed the strongest association with aspirin resistance with a cut-off of about 7 × 10^9^/L, which is within the normal range. Smoking history was another important factor associated with reduced aspirin effect.

## Figures and Tables

**Figure 1 pharmaceutics-14-02099-f001:**
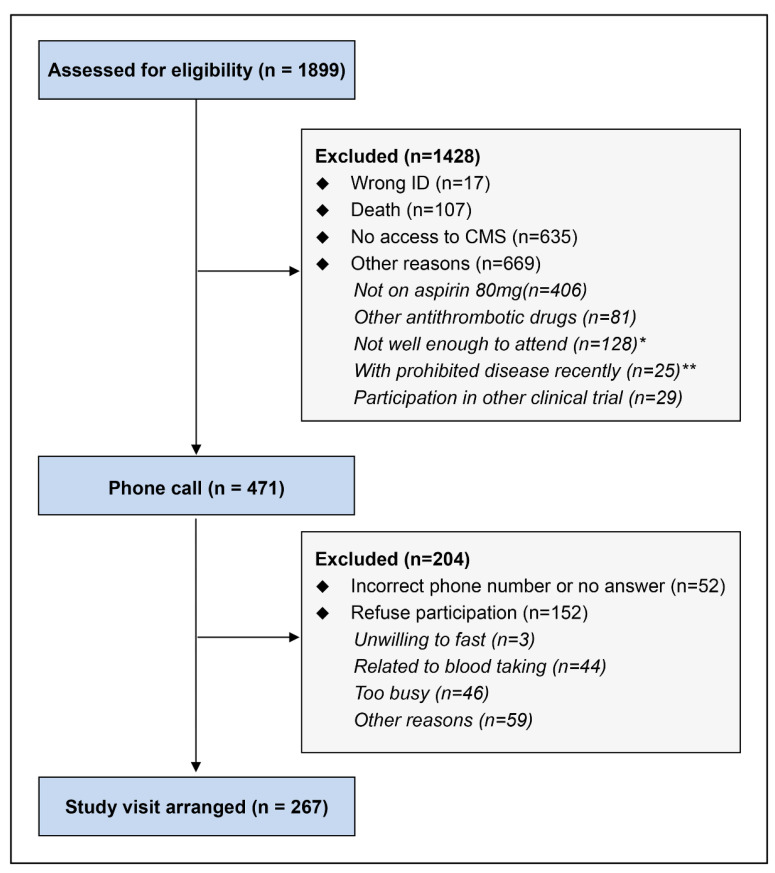
Patient flow diagram. * Patients with impaired mobility, or with dementia or hearing impairment. ** Some were excluded because of various other medical conditions.

**Figure 2 pharmaceutics-14-02099-f002:**
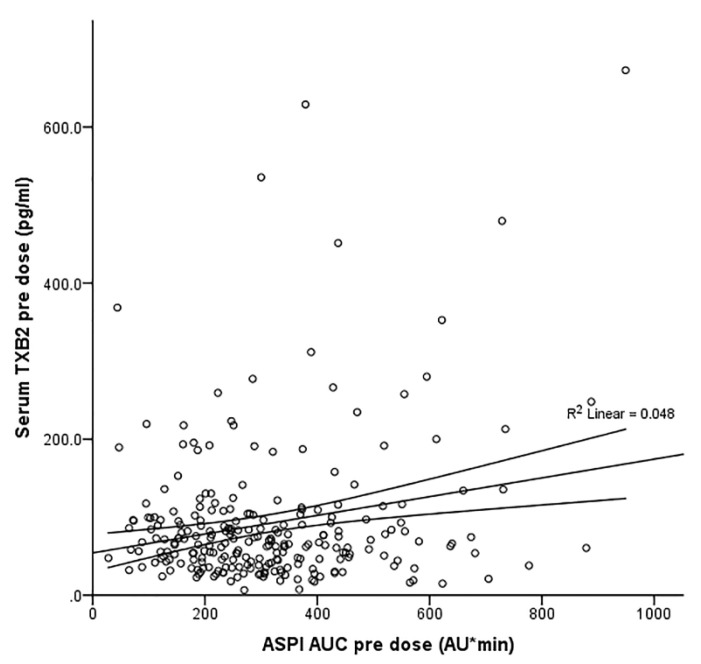
Correlation of ASPI AUC and serum TXB_2_ values pre dose after the adjustment for gender and BMI (*n* = 247, R = 0.219, R^2^ = 0.048, *p* = 0.001).

**Figure 3 pharmaceutics-14-02099-f003:**
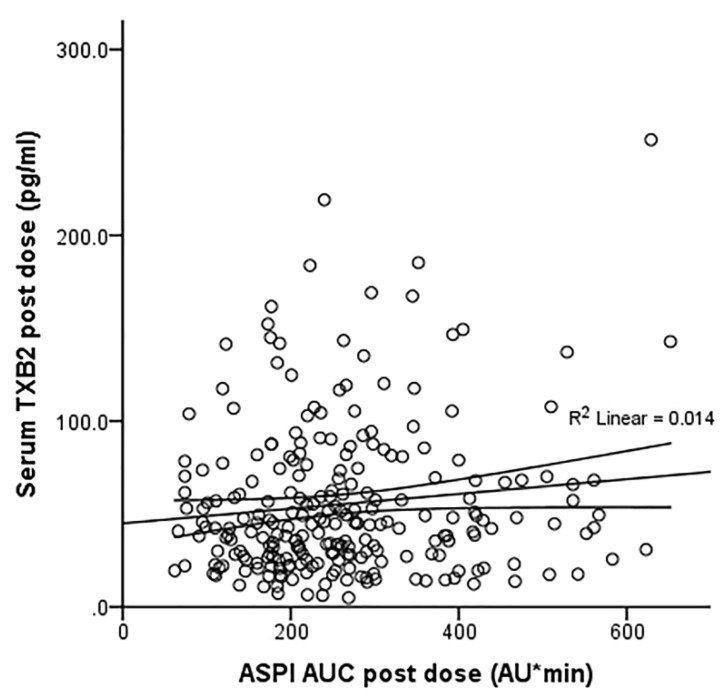
Correlation of ASPI AUC and serum TXB_2_ values 1 h post dose after adjustment for gender and BMI (*n* = 244, R = 0. 118, R^2^ = 0.014, *p* = 0.066).

**Figure 4 pharmaceutics-14-02099-f004:**
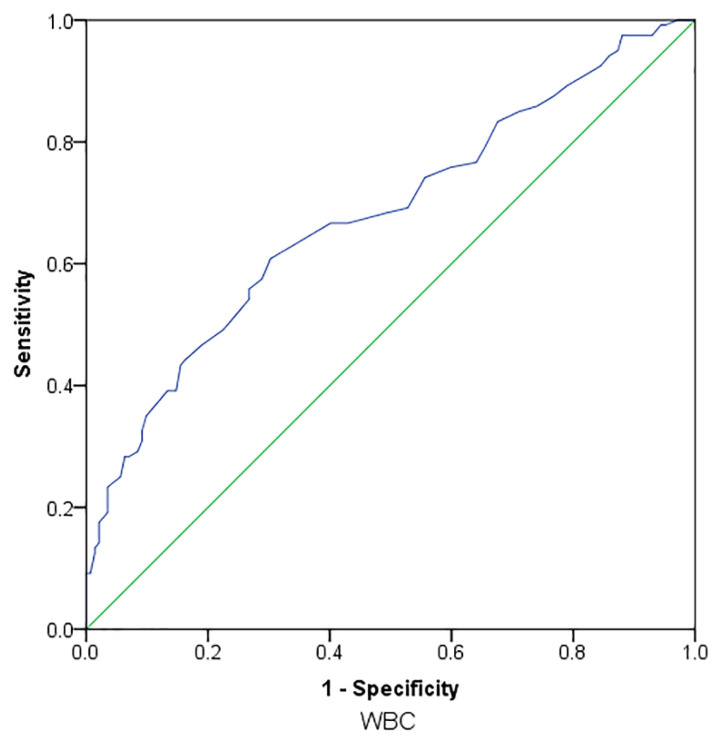
Receiver operating characteristic (ROC) plot of WBC count for aspirin resistance 24 h post dose.

**Figure 5 pharmaceutics-14-02099-f005:**
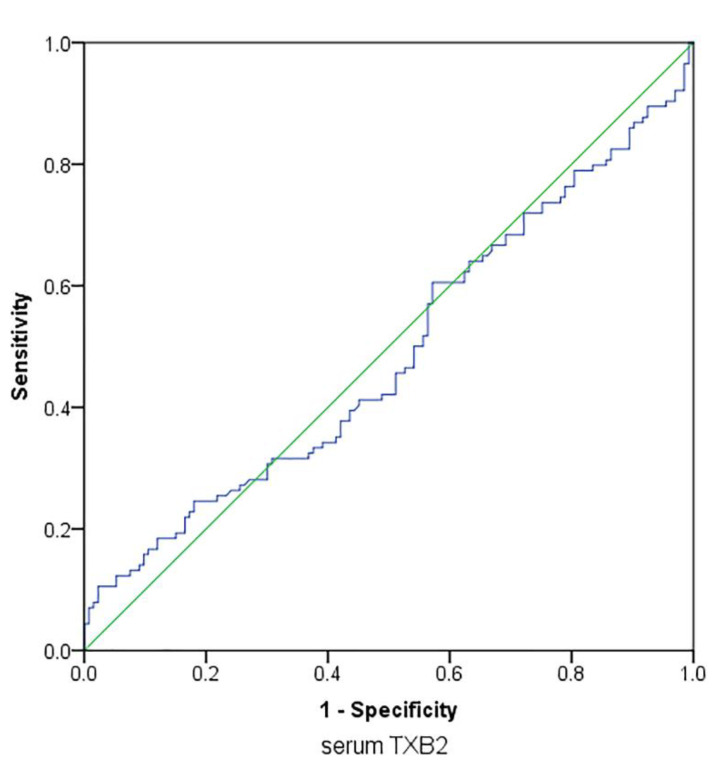
Receiver operating characteristic (ROC) plot of serum TXB_2_ for aspirin resistance after 24 h post dose.

**Table 1 pharmaceutics-14-02099-t001:** Clinical details of subjects divided into those with and without diabetes mellitus (DM).

Subjects	All		Non-DM		DM		*p*-Value
**Total, n**	**266**	**%**	**156**	**%**	**110**	**%**	
Female, n (%)	46	17%	24	15%	22	20%	NS
Prior CHD, n (%)	165	62%	95	61%	70	64%	NS
Prior angina, n (%)	253	95%	148	95%	105	95%	NS
Prior ACS, n (%)	151	57%	96	62%	55	50%	0.062
Prior PCI, n (%)	262	98%	153	98%	109	99%	NS
Prior ETT done, n (%)	67	25%	40	26%	27	25%	NS
PAD, n (%)	7	3%	2	1%	5	5%	0.06
Hypertension, n (%)	214	80%	114	73%	100	91%	<0.0001
Hyperlipidaemia, n (%)	263	99%	155	99%	108	98%	NS
Diabetes, n (%)	110	41%					
Smoker, n (%)	145	55%	91	58%	13	12%	NS
Stroke, n (%)	27	10%	9	6%	18	16%	0.005
Family history of CHD, n (%)	83	31%	51	33%	32	29%	NS
**Medication**							
HT Rx, total (%)	247	93%	142	91%	105	95%	NS
B-blocker, n (%)	194	73%	113	72%	81	74%	NS
ACEI/ARB, n (%)	165	62%	84	54%	81	74%	0.001
Statin, n (%)	256	96%	152	97%	104	95%	NS
DM Rx, total (%)	95	36%			95	86%	
Metformin, n (%)	81	30%			81	74%	
Insulin, n (%)	22	8%			22	20%	

Abbreviations: CHD, coronary heart disease; ACS, acute coronary syndrome; PCI, percutaneous coronary intervention; ETT, exercise tolerance test; PAD, peripheral artery disease; HT, hypertension; ACEI/ARB, previous stroke and to be treated with angiotensin converting enzyme inhibitor/angiotensin receptor blocker; NS, no significance.

**Table 2 pharmaceutics-14-02099-t002:** Demographic and laboratory details of subjects divided into those with and without diabetes mellitus (DM).

Demographics	All		Non-DM		DM		*p*-Value
	n	Mean ± SD	n	Mean ± SD	n	Mean ± SD	
Age, Years	266	66.6 ± 10.7	156	64.8 ± 11.5	110	69.1 ± 8.8	0.001
Body weight, kg	266	67.9 ± 11.7	156	67.7 ± 11.9	110	68.2 ± 11.5	NS
BMI	266	25.1 ± 3.5	156	24.9 ± 3.6	110	24.9 ± 3.6	NS
Body fat, %	266	25.1 ± 6.7	156	25.6 ± 7.0	110	24.4 ± 6.3	NS
Waist, cm	266	91.3 ± 9.9	156	90.2 ± 9.9	110	92.9 ± 9.8	0.03
Hip, cm	266	96.1 ± 6.4	156	96.1 ± 9.4	110	96.1 ± 6.4	NS
WHR	266	0.9 ± 0.1	156	0.9 ± 0.1	110	1.0 ± 0.1	0.002
SBP, mmHg	265	144.0 ± 22.5	156	140.0 ± 21.5	109	150.0 ± 22.8	0.001
DBP, mmHg	265	80.3 ± 9.5	156	81.1 ± 9.3	109	79.3 ± 9.8	NS
Pulse, beats/min	265	62.9 ± 8.3	156	62.4 ± 7.9	109	63.7 ± 8.9	NS
**Blood results**							
Fasting glucose,	266	6.2 ± 1.6	156	5.4 ± 0.6	110	7.3 ± 1.8	<0.0001
Total cholesterol,	266	3.9 ± 0.7	156	4.0 ± 0.7	110	3.8 ± 0.7	0.04
HDL-c,	266	1.3 ± 0.4	155	1.3 ± 0.4	109	1.2 ± 0.3	0.006
LDL-c,	266	2.0 ± 0.6	156	2.0 ± 0.6	110	1.9 ± 0.6	0.024
Triglyceride,	266	1.4 ± 0.9	148	1.3 ± 0.8	104	1.6 ± 1.0	0.007
non-HDL-C,	266	2.7 ± 0.7	57	2.6 ± 0.7	58	2.6 ± 0.7	NS
Urate,	115	0.4 ± 0.1	155	0.4 ± 0.1	110	0.4 ± 0.1	0.01
Platelet count,	265	202.5 ± 53.3	156	200.0 ± 50.7	110	206.6 ± 56.8	NS
MPV,	265	9.0 ± 0.9	156	9.0 ± 0.9	110	8.9 ± 0.8	NS
RBC,	266	4.7 ± 0.6	156	4.8 ± 0.5	110	4.5 ± 0.7	<0.0001
Hb,	266	13.7 ± 1.5	156	14.2 ± 1.2	110	13.1 ± 1.6	<0.0001
Haematocrit,	266	0.4 ± 0.1	156	0.4 ± 0.0	110	0.4 ± 0.1	<0.0001
WBC count,	266	6.6 ± 1.7	156	6.6 ± 1.7	110	6.7 ± 1.6	NS

Abbreviations: BMI, body mass index; WHR, waist-to-hip ratio; SBP, systolic blood pressure; DBP, diastolic blood pressure; HDL-c, high density lipoprotein cholesterol; LDL-c, low density lipoprotein cholesterol; MPV, mean platelet volume; RBC, red blood cell; Hb, haemoglobin; WBC, white blood cell; NS, no significance.

**Table 3 pharmaceutics-14-02099-t003:** Platelet aggregation parameters of subjects before and one hour after taking 80 mg aspirin of CHD patient with and without diabetes mellitus (DM).

	All			Non-DM			DM		
	n	Pre Dose	1 h Post Dose	n	Pre Dose	1 h Post Dose	n	Pre Dose	1 h Post Dose
Serum TXB_2_ (pg/mL)	251	91.3 ± 90.7	55.7 ± 40.4 ***	146	85.3 ± 86.8	52.2 ± 39.4 ^#^	105	99.6 ± 95.6	60.4 ± 41.4 ^#^
Area under the curve (AUC) (AU*min)	255	311.8 ±167.5	260.8 ± 122.1 ***	154	319.4 ± 167	266.6 ± 127.3 ^#^	108	300.9 ± 168.5	252.3 ± 114.1 ^#^

Values are given as mean ± SD, *** *p* < 0.001 compared between pre dose by paired *T*-test, # *p* < 0.0001 compared between pre dose by paired *T*-test.

**Table 4 pharmaceutics-14-02099-t004:** Multiple stepwise regression analysis for the correlates of the aggregation on the ASPI measurement 24 h post dose.

Parameters	Unstandardized Coefficients		Standardized Coefficients	t	*p*-Value
	B	SE	β		
Constant	−491.229	123.046		−3.992	0.000
WBC count	34.725	6.017	0.344	5.772	0.000
Hct	772.905	229.603	0.187	3.366	0.001
Heart rate	2.629	1.116	0.131	2.355	0.019
Platelet count	0.426	0.190	0.135	2.235	0.026
					R^2^ = 0.231

Abbreviations: WBC, white blood cell; Hct, haematocrit.

**Table 5 pharmaceutics-14-02099-t005:** Multiple stepwise regression analysis for the correlates of the aggregation on the ASPI measurement 1 h post dose.

Parameters	Unstandardized Coefficients		Standardized Coefficients	t	*p*-Value
	B	SE	β		
WBC count	21.730	4.304	0.296	5.049	0.000
Smoking history	43.894	14.362	0.179	3.056	0.002
Heart rate	1.918	0.856	0.129	2.241	0.026
					R^2^ = 0.163

Abbreviation: WBC, white blood cell.

## Data Availability

The data presented in this study are available on request from the corresponding author.

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
