# Peer review of "Factors Associated with Aspirin Resistance in Hong Kong Chinese Patients with Stable Coronary Heart Disease Using the Multiplate® Analyzer and Serum Thromboxane B2"

_pharmaceutics, 2022, doi:10.3390/pharmaceutics14102099_

Round 1

Reviewer 1 Report

Proposed paper is interesting and well written. However, some revisions are needed before paper can be accepted for publication:

- If DM at the end doesn't results as a predictor of resistance why many attention is payed to these in the previous table and patients were analyzed depending on the presence of DM? or definee this choose or delet these specific analisys.

- Hypertension have been associated to ASA resistance, please add it into the model.

- The cut-off values for WBC is fundamentally a normal values. In this sence its probably unsufull in the clinic because most of the patients will have values higher that the cut-off. I would remove this analysis. 

Author Response

1. Proposed paper is interesting and well written. However, some revisions are needed before paper can be accepted for publication:

Response: We would like to thank the reviewer for the very helpful comments.

2.If DM at the end doesn't results as a predictor of resistance why many attention is payed to these in the previous table and patients were analyzed depending on the presence of DM? or definee this choose or delet these specific analisys.

Response: Diabetes (DM) is one of the conditions that has frequently been associated with aspirin resistance in previous studies and we therefore considered that comparing patients with and without DM would be a useful approach. As the reviewer points out, our finding was that having DM was not associated with any of the measures of aspirin resistance in this study. However, as we took this approach from the start and there were several differences between the subjects with and without DM, as shown in tables 1 and 2, we would like to retain this part of the analysis if the reviewer can find this acceptable. We think it provides a useful message that DM is not always associated with aspirin resistance and we provided some discussion related to that.

3. Hypertension have been associated to ASA resistance, please add it into the model.

Response: We agree that hypertension maybe associating with aspirin resistance. In fact, we did include the systolic and diastolic blood pressures in the analysis. We did not include having or not having hypertension as a dichotomous variable. Many of the patients with a diagnosis of hypertension had blood pressure within the normal range at the time of the study visit and patients with hypertension were on various different treatments which may themselves have an effect on platelet aggregation, so we felt that adding hypertension based on the previous diagnosis as one of the factors in the multiple stepwise regression analysis may not be helpful. We added a sentence and a reference to the discussion to emphasize the association of hypertension and aspirin resistance.

“Aspirin resistance has also been associated with hypertension [38, 39] but we did not find any association with systolic and diastolic blood pressures in this study. As with diabetes, the treatments used for hypertension may interact with the effect of aspirin on platelets.”

4. The cut-off values for WBC is fundamentally a normal values. In this sence its probably unsufull in the clinic because most of the patients will have values higher that the cut-off. I would remove this analysis.

Response: Thank you for this comment. We agree that the cut off value for WBC identified by the ROC analysis is within the normal range and would not really be useful to identify patients with aspirin resistance. The high WBC value was the most significant finding in the multiple stepwise regression analysis and this has been reported to be associated with aspirin resistance in previous studies. We think it is worth mentioning so that clinicians may be aware that patients with WBC values in the upper part of the normal range or above the normal range maybe at risk of having aspirin resistance. We added an additional comment about this in the discussion. 

Reviewer 2 Report

This is an interesting premise evaluated in a prospective, observational study. However, my enthusiasm for this work is diminished by the emphasis on it being predictive in nature. There is guidance about the minimum sample size for validation of a clinical prediction model (https://doi.org/10.1002/sim.9025). There are several items in the current report that do not follow this guidance including: 1) the lack of external validation (either by dividing the cohort into derivation and validation cohorts or by having a second cohort) and b) by not using the principals used in the paper to help guide the sample size calculation.

If the authors are willing to drop the predictive language from their manuscript, then a discussion about the multivariable methods can be had. It appears that the authors utilized a stepwise logistic regression analysis for Tables 4 and 5, although this is not said in the methods section. There is also no explanation of why the authors did not consider a conceptual model for their regression analyses, which lets the computer determine the best mathematical fit regardless of whether there is any physiological basis for the covariates or not (or if there is collinearity impacting the analyses).

The relative merits of Table 4 vs Table 5 are also confusing to me. Are there any data suggesting that the trough or the peak value is more important in terms of the ASPI measurements?

Author Response

1. This is an interesting premise evaluated in a prospective, observational study. However, my enthusiasm for this work is diminished by the emphasis on it being predictive in nature. There is guidance about the minimum sample size for validation of a clinical prediction model (https://doi.org/10.1002/sim.9025). There are several items in the current report that do not follow this guidance including: 1) the lack of external validation (either by dividing the cohort into derivation and validation cohorts or by having a second cohort) and b) by not using the principals used in the paper to help guide the sample size calculation.

Response: We would like to thank the reviewer for the very helpful comments and for the advice to drop the predictive language from the manuscript. We agree with the reviewer and the useful reference provided, which we have now cited, that the study is not appropriate to provide a clinical prediction model and we should not have used that terminology. We have added a sentence along with the new reference to make this point.

2. If the authors are willing to drop the predictive language from their manuscript, then a discussion about the multivariable methods can be had. It appears that the authors utilized a stepwise logistic regression analysis for Tables 4 and 5, although this is not said in the methods section. There is also no explanation of why the authors did not consider a conceptual model for their regression analyses, which lets the computer determine the best mathematical fit regardless of whether there is any physiological basis for the covariates or not (or if there is collinearity impacting the analyses).

Response: We did use a stepwise logistic regression analysis to generate the data in tables 4 and 5. We have now explained that in the methods section. The regression diagnostics showed that the model studentized residuals of histogram followed a normal distribution and none of the included variables had collinearity (VIF < 10).

In retrospect, we could have used a conceptual model for the regression analysis and perhaps that would have been more appropriate. We adopted a similar approach to that used in some other studies (Larsen SB, Grove EL, Neergaard-Petersen S, Wurtz M, Hvas AM, Kristensen SD. Determinants of reduced antiplatelet effect of aspirin in patients with stable coronary artery disease. PLoS One 2015;10:e0126767).

3. The relative merits of Table 4 vs Table 5 are also confusing to me. Are there any data suggesting that the trough or the peak value is more important in terms of the ASPI measurements?

Response: We examined the trough and peak effects of the aspirin dosage in an attempt to identify whether it will be more useful to increase the frequency of dosing or the dosage amount for the aspirin. We also considered that the peak effect may be a more reliable indicator of the aspirin response as the aspirin intake was witnessed. We agree that it is not clear whether it is more important to examine the trough or peak effect, but we tend to think that the trough effect may be more important overall as it identifies more patients who appeared to have inadequate platelet inhibition.

Reviewer 3 Report

Nice work and useful results. 

This paper should be improved in Discussion. 

All the factors such as WBC, heart rate, and Hct should be explored as potential predictors and give more detailed mechanisms of their contributions to aspirin resistance. 

Limitations should be more detailed considering the sample size and participants' characteristics. 

Author Response

1. Nice work and useful results.

Response: Thank you for the kind comment.

2. This paper should be improved in Discussion. 

Response: Thank you for the helpful comment. We have added more sentences in the discussion.

3. All the factors such as WBC, heart rate, and Hct should be explored as potential predictors and give more detailed mechanisms of their contributions to aspirin resistance.

Response: We have added more detail in the discussion about the potential mechanisms for these factors to be associated with aspirin resistance. 

4. Limitations should be more detailed considering the sample size and participants' characteristics. 

Response: We have added more details about the limitations, particularly regarding the small sample size. And we have added references 37 to 42 in the manuscript.

Round 2

Reviewer 1 Report

Authors replies to all the query raised and paper can now be accepted for publication.

Author Response

1. Authors replies to all the query raised and paper can now be accepted for publication.

Response:Thanks.

Reviewer 2 Report

Thank you for your responsiveness to my feedback. I appreciate the changes made by the authors.

1. Predicting is still mentioned in the abstract for WBCs. Also, smoking history as a "predictor" in the conclusion section. Please revise.

Author Response

1. Predicting is still mentioned in the abstract for WBCs. Also, smoking history as a "predictor" in the conclusion section. Please revise.

Response: Thank you very much for your very careful review of the revised manuscript. We apologise for not making those changes before. We thought we had removed all reference to prediction for this study. We have now changed those two mentions, thank you.